# Family Caregiving during the COVID-19 Pandemic in Canada: A Mediation Analysis

**DOI:** 10.3390/ijerph19148636

**Published:** 2022-07-15

**Authors:** Sharon Anderson, Jasneet Parmar, Tanya L’Heureux, Bonnie Dobbs, Lesley Charles, Peter George J. Tian

**Affiliations:** 1Division of Care of the Elderly, Department of Family Medicine, University of Alberta, Edmonton, AB T5G 2T4, Canada; jasneet.parmar@ahs.ca (J.P.); tanyarlheureux@gmail.com (T.L.); bdobbs@ualberta.ca (B.D.); lesley.charles@albertahealthservices.ca (L.C.); peter.tian@ualberta.ca (P.G.J.T.); 2Medically At-Risk Driver Centre, University of Alberta, Edmonton, AB T5G 2T4, Canada; 3Alberta Health Services, Edmonton, AB T5G 0B7, Canada

**Keywords:** family caregivers, carers, anxiety, frailty, loneliness, social support, COVID-19, mediation analysis

## Abstract

Family caregiving is a public health issue because of caregivers’ significant contribution to the health and social care systems, as well as the substantial impact that giving and receiving care has on the health and quality of life of care receivers and caregivers. While there have been many studies that associate caregivers’ care work, financial difficulty, navigation, and other caregiving factors with family caregivers’ psychological distress, we were interested not only in the factors related to family caregiver anxiety but also in hypothesizing how those effects occur. In this study, we used Andrew Hayes’ PROCESS moderation analysis to explore the link between caregiver frailty, weekly care hours, and perceptions of financial difficulty, social support, and anxiety. In this analysis, we included 474 caregivers with relatively complete data on all of the variables. In regression analysis after controlling for gender and age, social loneliness (β = 0.245), frailty (β = 0.199), financial difficulty (β = 0.196), care time (β = 0.143), and navigation confidence (β = 0.131) were all significant. We then used PROCESS Model 6 to determine the significance of the direct, indirect, and total effects through the serial mediation model. The model pathway from frailty to care time to financial difficulty to social loneliness to anxiety was significant. The proportions of family caregivers who were moderately frail, anxious, and experiencing social loneliness after eighteen months of the COVID-19 pandemic found in this survey should be of concern to policymakers and healthcare providers.

## 1. Introduction

Public health’s goal is to protect and improve the health of individuals, communities, and populations of all sizes from neighborhoods to the world. Family caregiving is a public health issue because of caregivers’ significant contribution to the health and social care systems within countries [1,2,3], as well as the substantial impact that giving and receiving care has on the health and quality of life of care-receivers and caregivers [4,5].

Current Canadian health systems depend on family caregivers to provide 80–90% of the day-to-day assistance and care management required by care-receivers living in the community [6,7], and assist with 15 to 30% of the care for congregate-care residents [8,9,10]. Having a caregiver is associated with care-receiver’s decreased healthcare utilization and risk of institutionalization [11]. Many family caregivers are caring more intensively and for longer due to increased life expectancies, a higher proportion of older adults living with frailty and complex chronic conditions who need care, and structural changes such as smaller families, divorce, geographic mobility, and employed spouses/partners that have reduced family caregiver supply [12,13,14]. Caregivers are experiencing increasing frailty [15], deteriorating physical health [14], declining levels of cognitive functioning [16], and increasing distress and anxiety [17,18,19,20]. In 2010, 16% of Canadian caregivers of home care clients were distressed [7] and distress rose to 33% by 2016 [21,22]. Anxiety rates of over 50% were reported during the COVID-19 pandemic [19,23,24,25,26].

Exposure to chronic stress and anxiety has been proposed as accelerators for caregivers’ health decline [26,27] and frailty [28,29]. However, frailty and poor health can also increase caregiver stress [28,30]. For example, Smagula and colleagues [30] showed that 82% of caregivers providing intensive help with activities of daily living were suffering from anxiety; however, those who were frail, that is, exhibiting white matter brain pathology, reported strain at much lower levels of care work.

### Review: Factors Associated with Caregiver Anxiety

Caregiving intensity, whether it is measured by hours of care or the type and the quantity of assistance provided, is associated with caregiver anxiety and distress [6,13]. InterRAI Home Care assessment research associates caregiver distress with caring for more than 21 hours weekly, caring for someone with dementia, depression, or dementia-related responsive behaviors, and co-residing with the care-receiver [31,32,33]. Distress on RAI-Home Care assessment includes expressing feelings of anger, depression, or the inability to continue with caring activities.

Family caregivers reported that their care work increased substantially in the COVID-19 pandemic [23,24,34]. Objective assessments confirmed that formal support for family caregivers in the first six months of the COVID-19 pandemic (March–September 2020) decreased [25]. In Canada, Sinn and colleagues [34] found there were significantly fewer home care admissions, significantly fewer standardized assessments, and significantly more clients who received no personal support services or rehabilitation services despite assessed needs. Clients who had been receiving services received significantly fewer hours of personal support and therapy visits per month.

Out-of-pocket caregiving costs [35] and difficulty navigating health and community systems [33,36,37,38] are also identified as factors that contribute to family caregiver workload and anxiety [33,36,37,38]. In 2012, one in five Canadian caregivers were experiencing financial hardship [36]. Caregivers experience financial hardships from out-of-pocket care expenses as well as lost income and reduced pensions due to reduced work hours or leaving the workforce to care. In 2017, Taylor and Quesnel-Vallée [38] estimated that family caregivers were spending 15 to 50% of the time on the structural burden of care—that is finding, negotiating for, and then managing health and social care services for the care receiver.

Social supports can moderate the effect that stressors like intensive care work and financial difficulty have on family caregivers [20,39,40]. According to stress-buffering theory, perceptions that social support is available, that is, family or friends who one feels close to or are available to help when one has a problem, can mitigate the effects of stressors on caregiver’s anxiety and distress [41,42]. During the pandemic, social support from family and friends decreased [23,24,43,44,45].

While there have been many studies that associate care work, financial difficulty, and inadequate social support with increases in family caregivers’ psychological distress, little research has been conducted on how these factors might connect to caregiver health and anxiety. In this study, we used Andrew Hayes’ PROCESS moderation analysis to explore the link between caregiver frailty, weekly care hours, perceptions of financial difficulty, and social support and anxiety.

Theoretically, we hypothesized that: (1) caregivers’ health (frailty) is positively associated with anxiety; (2) the association between caregiver frailty and caregiver anxiety will be mediated by weekly care work, perceived financial difficulty and social support; and (3) there will be a causal serial mediation pathway from care time to perceived financial difficulty and social support. The proposed serial mediation model in this study is depicted in Figure 1.

## 2. Materials and Methods

A cross-sectional survey of the effects of COVID-19 on Alberta family caregivers was conducted online on the secure REDCap survey platform from 21 June 2021 to 31 August 2021. Alberta is a land-locked western province in Canada. About 15% of the population lives in rural and remote areas and 85% lives in urban and suburban settings. The inclusion criteria were: (1) 14 years of age and older; (2) a family caregiver (carer, care partner) defined as any person who takes on a generally unpaid caring role providing emotional, physical, or practical support in response to another person’s disability, mental illness, drug or alcohol dependency, chronic condition, dementia, terminal or serious illness, frailty from ageing, or COVID-19 [46]; and (3) resides in Alberta.

The health research ethics board at the University of Alberta approved all study methods. To recruit participants, we approached health and community organizations who work with family caregivers by email to advise family caregivers about the survey in their newsletters and posters. We also used social media platforms such as Twitter, LinkedIn, and Facebook to inform family caregivers directly about the survey. All participants were asked to read information about the study and provided implied consent by clicking on the survey (Appendix A). Of the 685 people who clicked on the survey, 556 current family caregivers responded to more than three quarters of the questions (81.17%).

### 2.1. Data Collection

The survey consisted of 30 closed questions (Likert scale, yes/no, list) and five open-ended questions. As per our ethics approval, participants were informed it was their choice to answer or skip questions they did not wish to answer. The survey sections used in this study consisted of four main sections: (1) care work; (2) health (frailty, anxiety, changes in physical and mental health); (3) stressors (financial, navigation, social loneliness); and (4) demographics (of both caregiver and care-receiver). The full survey can be found in the supplemental materials (Appendix A).

#### 2.1.1. Weekly Care Work

Family caregivers who were caring before COVID-19 were asked whether care time increased, remained stable, or decreased. We assessed the number of hours devoted to weekly care time during COVID-19 with the following options ≤10 h, 11–20 h, 21–30 h, 31–40 h, 41–80 h, 81–120 h and 121–168 h.

#### 2.1.2. Frailty

Frailty has been acknowledged as a new public health priority because of its association with multimorbity, hospitalizations, hospital readmission, institutionalization, increased healthcare costs and mortality [47,48,49]. Rockwood [49] defines frailty “as the term widely used to denote a multidimensional syndrome of loss of reserves (energy, physical ability, cognition, health) that gives rise to vulnerability.” The primary reason for congregate care admission is the caregiver’s health failing where they become too frail to care. We consider frailty screening to be one step in an approach towards promoting family caregivers’ health and wellbeing. In this study we used a self-report version of the Clinical Frailty Scale [CFS] [48,49], initially used in an assessment study of the effects of frailty assessment and social prescribing [50]. The CFS was originally validated for face-to-face screening by a health provider, but adaptations are permitted. We used Rasiah’s [50] nine questions to assess caregiver’s frailty (See CFS Scale in Appendix A). 

#### 2.1.3. Anxiety

We assessed anxiety with Tluczek and colleagues’ [51] validated [51,52,53] six-item State Anxiety Scale short-form version of the State Trait Anxiety Inventory [STAI]. Both the long and short forms measure feelings of worry, tension, nervousness, and apprehension, with questions like “I feel comfortable” or “I feel good”. Questions 1, 3, and 6 are worded positively, then reversed scored so that higher scores indicate higher anxiety. The score is multiplied by 20/6 to obtain scores ranging from 20–80. The Cronbach’s alphas for the six-item short-form State Anxiety Scale ranged from 0.74 to 0.82 [51,52,53]. In this case, the standardized Cronbach’s alpha was 0.85.

#### 2.1.4. Social Loneliness

In this study we used the three item social loneliness subscale from the DeJong Gierveld Loneliness Scale [54] to measure caregivers’ perceptions of the social support available from family and friends. The three positively worded questions “There are plenty of people I can lean on in case of trouble”, “There are plenty of people I can count on completely”, and “There are enough people I feel close to” are answered on a three-point scale of no, more or less, and yes. To avoid bias, “no” and “more or less” are scored as 1, and “yes” as a 0, yielding total scores of 3. Higher scores indicate more social loneliness or lower social support. In a large seven country sample, the social loneliness subscale had a 0.85 Cronbach’s alpha [54], and in this provincial sample, the standardized alpha coefficient was 0.89.

#### 2.1.5. Other Stressors

We assumed that financial difficulty and difficulty navigating health and social care systems could be stressful. To assess caregivers’ perceptions of their financial difficulty, we asked them, “During the pandemic, have you experienced financial hardships because of caregiving responsibilities?” rated on a scale of none, a few, moderate, and a lot. The ability to access services and supports was assessed with the question, “How would you rate your ability to access services and navigate the healthcare system?” with answers “very capable”, “somewhat capable”, “neutral”, “a little capable”, and “not at all capable”.

### 2.2. Data Analysis

Data analysis proceeded in three stages. First, we used descriptive statistics to examine distributions for categorical variables and means and standard deviations for continuous variables. We explored associations with Pearson correlation analysis stepwise linear regression to examine the caregiver factors associated with caregivers’ anxiety even though Hayes advises that the moderator variables significant in the PROCESS moderation analysis may not be significant in regression models [55]. A three-step hierarchical linear regression was conducted with anxiety as the dependent variable. The hierarchical regression analysis results consist of model comparisons and a model interpretation based on an alpha of 0.05. Then, in order to address our moderation hypothesis, we used SPSS Version 26 (IBM, Armonk, NY, USA) and version 4 of Hayes [55] PROCESS moderation/ mediation data analysis program. The answers “Don’t know” and “Prefer not to answer” were treated as missing values and were managed by excluding them list-wise. The statistical significance was set at *p* < 0.05.

The SPSS PROCESS model runs each predictor factor individually. The moderation models were tested in two steps, first the parallel model and then the serial process version. Both the parallel (Model 4) and serial (Model 6) moderation analysis based on 5000 bootstrap samples using a 95% confidence interval were calculated.

## 3. Results

### 3.1. Participants

In this analysis, we included 474 caregivers with relatively completed data on all of the variables. The majority were women (83.3%), were 55 to 64 years of age (36.5%), cared for a parent (43.9%), and were well-educated, that is had college or technical training and higher (86.4%). Over two-thirds (66.7%) rated themselves as healthy (1–3 on the frailty scale). Almost half (48.7%) were experiencing financial difficulty, 44.3% were providing care for 21 or more hours per week, and 13.4% were not confident about their ability to navigate health and community systems (See Table 1).

### 3.2. Caregiver Factors Associated with Caregiver’s Anxiety

Table 2 summarizes the hierarchal linear regression models testing the independent associations between demographics, stressors, and perceived social support (social loneliness). Multicollinearity was not present (tolerance values were more than 0.1 and variance inflation factors were less than 10). See Appendix A: Appendix A Scatterplot with Loess curve to check homoscedasticity assumptions; Appendix A: Appendix A P-P Plot, checking for normality of estimation error assumption; and Appendix A: Appendix A Correlations among key variables.

In step one, age and gender explained 4.6% of the variance in anxiety. Stressors, including weekly hours of care work, difficulty navigating, financial difficulty, and caregiver frailty explained an additional 25.3% of the variance in anxiety in step 2. In step 3, social loneliness explained an additional 5.3% of the variance in anxiety. In this final model, all the factors remained significantly and independently related to worse caregiver anxiety and explained 35.2% of the variance in anxiety.

### 3.3. Testing the Parallel Mediation Model

Theoretically, we assumed that the direct effect of caregiver frailty on anxiety was mediated by weekly hours of care work, financial difficulty, and social loneliness. After controlling for age, gender, and navigation confidence, we used PROCESS Model 4 to determine the significance of the direct, indirect and total effects of a parallel mediation model. Results of the parallel mediation analysis indicate that frailty is indirectly related to anxiety through the relationships with care time, financial difficulty, and social loneliness. As can be seen in Figure 2 and Table 3, caregivers reporting more frailty reported higher levels of anxiety (β = 2.159, t = 4.909, 0.0005), more care time (β = 0.570, t = 6.302, 0.0005), financial difficulty (β = 0.229, t = 6.79, 0.0005), and social loneliness (β = 0.269, t = 6.525, 0.0005). The mediators care time (β = 0.676, t = 3.148, 0.0017), financial difficulty (β = 2.673, t = 4.635, 0.0005), and social loneliness (β = 2.756, t = 6.076, 0.0005) were also positively associated with anxiety. On the bias-correct bootstrap analysis, the total indirect effects of the mediators on care work, financial difficulty, and social loneliness (a_1_ + b_1_ + a_2_ + b_2_ + a_3_ + b_3_ = 1.743, 95% CI 1.226, 2.351) were significant, agreeing with our parallel mediation hypothesis. See Table 3.

### 3.4. Testing the Serial Mediation Analysis

The serial mediation model assumes that there is a causal path from M_1_ care time to M_2_ financial difficulty to M_3_ social loneliness, and estimates this effect, whereas the parallel model assumes the effect is zero. After controlling for age, gender, and navigation confidence, we used PROCESS Model 6 to determine the significance of the direct, indirect and total effects of the serial mediation model. See Table 4. The total effect consists of a direct effect c’, from *X* frailty to *Y* anxiety and seven specific indirect effects (Figure 3). The direct effect c’ of frailty (X) on the anxiety (Y) is 2.16. The effect is significant (*p* = 0.0005). In other words, when controlling for other variables in the model, anxiety will increase by 2.16 points for a 1-point difference on the frailty scale.

Calculating the indirect effects requires computing the model pathway from frailty to care time (M1) to financial difficulty (M2) to social loneliness (M3) to anxiety. The 95% bias corrected confidence interval based on 5000 bootstrap samples indicated the indirect effects (a_1_ + b_1_ + a_2_ + b_2_ + a_3_ + b_3_ = 1.743, SE 0.297) to be entirely above zero (1.209, 2.369). Following Hayes [55], it was significant as the bootstrapping 95% confidence interval does not contain 0. This result lends support to the serial mediating role of care time, financial difficulty, and social loneliness in increasing family caregiver anxiety. Notably the serial mediation model pathway frailty to care time (M1) to social loneliness (M3) to anxiety that did not include financial difficulty (M2) was not significant (β = 0.056, SE 0.036, CI-0.011, 0.131). Moreover, caregivers with greater frailty reported greater anxiety even after taking into account frailty’s indirect effects through care time, financial distress, and social loneliness (c’= 2.159, *p* = 0.0005).

## 4. Discussion

In this analysis, we were interested not only in the factors related to family caregiver anxiety, but also in hypothesizing *how* those effects occur. In regression analysis after controlling for gender and age, social loneliness (β = 0.245), frailty (β = 0.199), financial difficulty (β = 0.196), weekly care time (β = 0.143), and navigation confidence (β = 0.131) were all significant. We then used mediation analysis to evaluate the hypothesis that caregiver’s health measured as frailty transmitted its effect on anxiety during the COVID-19 pandemic.

Analysis confirmed our hypothesis that increasing weekly care time would likely increase financial difficulty which in turn would lead to social loneliness, and together positively mediated the effect of caregiver frailty on caregiver anxiety. In the serial mediation analysis, two caregivers that differ by one scale point in frailty are estimated to differ by 3.90 units in anxiety, with the frailer caregivers reporting higher anxiety. They differ by 1.74 units in anxiety as a result of the positive effect of frailty through weekly care work, which in turn is associated with more financial difficulty, social loneliness and anxiety. Independent of the mechanism of the indirect effects of weekly care work, financial difficulty, and social loneliness, the two caregivers are estimated to differ by 2.12 units in anxiety, with the more stressed frail caregiver reporting higher anxiety.

Smagula’s 2017 research [30] makes the link between increased anxiety from caregiving and the caregiver’s poor health measured in terms of white matter damage, yet caregiver frailty is often not considered in the caregiving research. In this study, about a third of caregivers rated themselves as moderately frail, that is, more tired, having more trouble obtaining supports, and/or needing practical or physical assistance with finances, transportation, or heavy housework (See scale in supplementary materials) [50]. Frailty is a public health measure that indicates increased risk of poor health [56,57,58]. Rockwood’s [56] notion in frailty is that as problems with health accumulate, they start to erode higher order functions like being able to “think and do as they please; look after themselves, interact with other people, and move about without falling” (p. 254).

We hypothesized that caregiver frailty, characterized by a decline in functioning across multiple physiological symptoms [49], would make it more exhausting to provide care as well as to coordinate and manage care. Numerous studies associate caregiver health with caregiver burden and distress [13,59,60,61]. Many authors propose that caregiving can adversely affect caregivers’ psychological and physical health [62,63,64,65], and that poor health will also increase distress [66,67,68,69,70,71]. Notably, in our serial mediation model, caregivers with greater frailty reported greater anxiety even after taking into account frailty’s indirect effect through care time, financial distress, and social loneliness.

Frail caregivers would benefit from a comprehensive frailty assessment and person-centered preventative approaches to improve their wellbeing [29,50]. In this survey we used the self-assessment version of the Clinical Frailty Scale [50] piloted to understand if assessing people’s risks and needs, and then facilitating referrals to healthcare and community resources could remediate frailty. Although identification and treatment of frailty is not currently standard practice in primary care [62], there is mounting evidence that assessing frailty in primary care and then person-centered intervention is feasible [72,73,74,75,76,77]. Frail caregivers would benefit from a comprehensive frailty assessment and person-centered preventative approaches to improve their wellbeing [29,50]. 

It is not surprising that weekly hours of care time mediated anxiety in this analysis. Caregiver distress begins to rise significantly for those providing over 21 hours of care per week [31,32]. Weekly care time and care intensity increased substantially at the outset of the COVID-19 pandemic as home care supports and respite were reduced, and day programs closed in order to redirect healthcare resources into acute care and reduce risk of COVID transmission [34,35,78]. Caregivers reported that without stimulation and social interaction, the care-receiver’s health deteriorated, which increased their care time and intensity. COVID-19 sanitation protocols also increased care time. In this survey, over half of the caregivers reported that their physical health (58%) and mental health (69%) deteriorated. Family caregivers would benefit from being asked about their situations by healthcare providers and access to more homecare supports.

Just under half (47.7%) of these caregivers reported that they were experiencing financial hardships due to their caregiving. Keating and colleagues’ taxonomy of care costs revealed three cost domains: employment consequences, care work, and care-related out of pocket costs [39]. In 2014, compared to before they started caring, Carers UK reported that caregivers experienced higher utility bills (77%), transportation costs (67%) and also spent more on cleaning products, food and clothing [79]. Qualitatively, in this survey, many caregivers reported they had to reduce their work hours or quit work to provide care.

We also asked caregivers what care expenses were higher since the COVID-19 pandemic was declared. They reported that during the COVID-19 pandemic they experienced higher food costs (44%); personal protective equipment and supplies (40%); care supplies such as incontinence products, meal replacements, bandages (29%); over the counter medical expenses (26%); mobility equipment like walkers and wheelchairs (17%); and household expenses (utility bills, rent, taxes, insurance) (24%). Extra expenses and financial distress can result in caregivers having to reduce their own expenses [39]. In addition to reducing their opportunities to save or invest, they may have to reduce spending on discretionary activities that bring them joy [39]. Poverty is the single largest determinant of health [80], and ill health is an obstacle to sustaining care [13,15,16]. We need to raise awareness of the effects of caregiving on incomes, employment, and pension credits.

Loneliness and social isolation also increased as the result of public health physical and social distancing protocols [23,24,81]. Our findings indicate that the association between frailty and loneliness was also moderated by social loneliness. In this survey the proportions of caregivers reporting that their social networks were lacking was high: 74% did not have enough people to rely on when they have problems, 73% did not have enough people they could trust completely, and 66% did not have enough people they felt close to. Social support from family and friends that typically is associated with reduced anxiety [40,41] was not available in the COVID-19 pandemic. Loneliness carries the same risk to health as smoking 15 cigarettes a day and being an alcoholic [82], thus ensuring that family caregivers have time to maintain their social networks should be a public health priority.

### Strengths and Limitations

This study has several limitations. First, we cannot equivocally claim that weekly care time, financial difficulty, and social loneliness cause anxiety. This is a cross-sectional study. We are basing our analysis on theoretical arguments which Andrew Hayes [55] suggests does not explicitly establish cause, stating that “one can conduct a mediation analysis even if one cannot unequivocally establish causality” (pg. 81). Second, these measures were all self-reported. The frailty measure, in particular, may be prone to a social desirability bias. In this survey the ratings went from 1, the least frailty, to 9, the most frailty. Rasaih and colleagues found that older people rated frailty higher when ratings went from most to least frailty [50]. Third, while anxiety and social loneliness were rated on well validated scales and were thus less subject to desirability bias, they are still self-report tools. They cannot be considered as reliable as objective diagnostic tools.

Despite these limitations, there are strengths. This study mirrors many other studies that demonstrated worldwide increases in caregiver anxiety, care work, financial difficulty, loneliness, and care work [23,24,25,26,43,78,81,83,84]. We used regression analysis and validated scales to examine the factors related to anxiety and then used mediation analysis to test hypotheses about the processes by which giving care can cause anxiety. Typically, anxiety rises as care responsibilities, exhaustion, and worry increase [67], and in this study, moderation analysis confirmed that increased weekly care hours, financial distress, and social loneliness were significant factors. While anxiety is a known risk factor for poor health [67], mediation analysis demonstrated that frailty also increases anxiety directly and indirectly. This implies that healthcare providers should routinely assess family caregivers’ frailty and distress.

## 5. Conclusions

Our findings provide some insight into the relationship between caregivers’ health, measured as frailty, and their anxiety. Frailty levels were directly related to levels of caregivers’ anxiety and also indirectly moderated through weekly hours of care work, financial distress, and social loneliness. The proportions of family caregivers who were moderately frail, anxious, and experiencing social loneliness after eighteen months of the COVID-19 pandemic found in this survey should be of concern to policy makers and healthcare providers.

## Figures and Tables

**Figure 1 ijerph-19-08636-f001:**
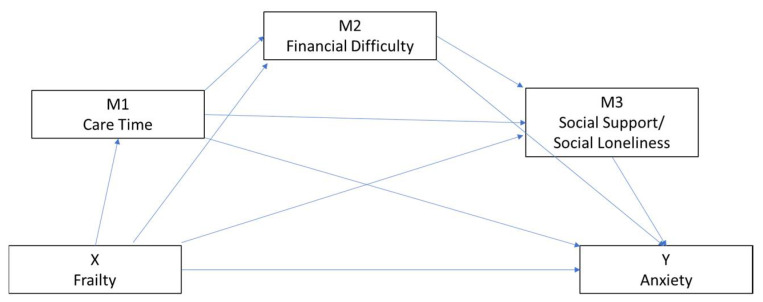
Hypothesized serial moderation model.

**Figure 2 ijerph-19-08636-f002:**
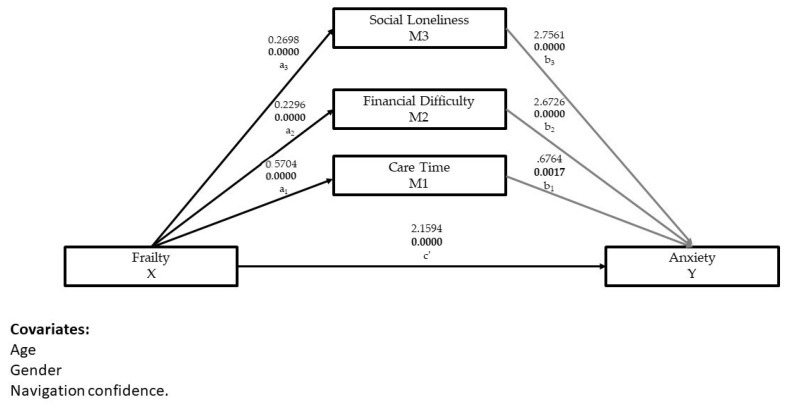
The parallel mediating effects of care time, financial difficulty, and social loneliness in a relationship between frailty and anxiety.

**Figure 3 ijerph-19-08636-f003:**
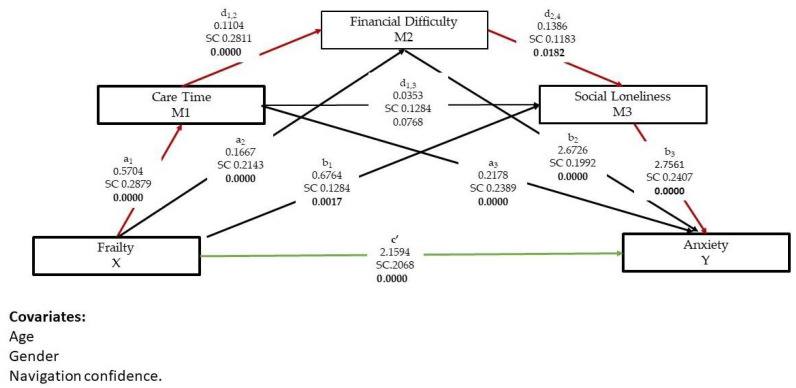
The serial mediating effects of care time, financial difficulty and social loneliness in a relationship between frailty and anxiety.

**Table 1 ijerph-19-08636-t001:** Caregiver socio-demographic variables.

Variables and Values	N (%)	M (SD)
**Sex**		
Women	396 (83.5%)	
Men	71 (15.0%)	
Non-Binary, Trans, Other	7 (1.5%)	
**Age**		
15–24	7 (1.5%)	
25–34	22 (4.6%)	
35–44	37 (7.8%)	
45–54	82 (17.3%)	
55–64	173 (36.5%)	
65–74	119 (25.1%)	
75–84	32 (6.8%)	
85–94	2 (0.4%)	
**Relationship to receiver**		
Parent/In-Law	208 (43.9%)	
Spouse/Partner	91 (19.2%)	
Sibling	33 (7.0%)	
Child	99 (20.9%)	
Other Relative	11 (2.3%)	
Friend/Neighbour	12 (2.5%)	
Other	20 (4.2%)	
**Education**		
Grade school	2 (0.4%)	
High school	63 (13.3%)	
College/ Technical training	175 (36.9%)	
University degree	121 (25.5%)	
Postgraduate degree, professional designation	110 (23.2%)	
**Weekly Care Hours**		
≤10	185 (39.0%)	
11–20	78 (16.4%)	
21–30	34 (7.2%)	
31–40	32 (6.8%)	
41–80	43 (9.1%)	
81–120	49 (10.5%)	
121–168	52 (11.0%)	
**Financial hardship**		
None	243(51.3%)	
A Few	130 (27.4%)	
Moderate	50 (10.5%)	
A Lot	51 (10.7%)	
**Navigation confidence**		
Very confident	162 (34.2%)	
Confident	188 (39.7%)	
Neutral	56 (11.8%)	
A little confident	46 (9.7%)	
Not at all confident	22 (4.6%)	
**Frailty (1–9)**		2.61 (1.32)
1–3 Good health	321 (67.7%)	
4–6 Frail	152 (32.1%)	
7–9 Severe Frailty	1 (0.2%)	
**Physical health changes in last year**		
Improved	19 (4%)	
Remained same	183 (39%)	
Deteriorated	272 (57%)	
**Mental health changes in last year**		
Improved	16 (3%)	
Remained same	132 (28%)	
Deteriorated	36 (69%)	
**Anxiety (20–80)**		48.45 (13.51)
≤41 Low anxiety	133 (28.1%)	
>42 Moderate-high	341 (71.9%)	
**Social Loneliness (0–3)**		2.13 (1.17)

**Table 2 ijerph-19-08636-t002:** Hierarchical Linear Regressions Results of Caregiver Factors with Caregivers’ Anxiety.

	Standardized β	95% CI	*p*
**Demographic factors**	R^2^ = 0.046 F = 11.33 (2,471), *p* < 0.001		
Age	−0.213	−3.15, −1.28	**<0.001**
Gender	−0.060	−4.80, 0.93	0.185
**Stressors**	R^2^ = 0.308 F = 44.11 (4,467), *p* < 0.001		
Age	−0.116	−20.3, −0.38	**0.004**
Gender	−0.0922	−5.44, −0.47	**0.039**
Care work: hours weekly	0.144	0.58,2.12	**<0.001**
Navigation confidence	0.122	0.52, 2.43	**<0.003**
Financial Difficulty	0.229	1.90, 4.24	**<0.001**
Frailty	0.268	1.93, 3.67	**<0.001**
**Perceived Social Support**	R^2^ = 0.359 F = 37.12 (1,466), *p* < 0.001		
Age	−0 0.117	−2.02, −0.42	**0.003**
Gender	−0 0.098	−5.53, −0.74	**0.010**
Care work: hours weekly	0.127	0.45,1.93	**0.002**
Navigation confidence	0.113	0.43, 2.28	**0.004**
Financial Difficulty	0.200	1.55, 3.81	**<0.001**
Frailty	0.210	1.33, 3.05	**<0.001**
Social Loneliness	0.241	1.87, 3.65	**<0.001**

**Table 3 ijerph-19-08636-t003:** Testing weekly care work, financial difficulty, and social loneliness as mediators in the relationship between caregiver frailty and anxiety.

Model Pathways		Coeff. β	Stan. Coeff.	SE	95% CI	*p*	Model Summary
**Total effect of X Frailty on Y Anxiety**	**c**	**3.903**		**0.439**	**3.04, 4.76**	**0.0005**	
*X* Frailty to M1 Care work	a_1_	0.570	0.2879	0.091	0.393, 0.748	0.0005	R^2^ = 0.088 F(4,469) = 11.368 *p* = 0.0005
*X* Frailty to M_2_ Financial difficulty	a_2_	0.229	0.2952	0.034	0.163, 0.296	0.0005	R^2^ = 0.174 F(4,469) = 24.651 *p* = 0.0005
X Frailty to M_3_ Social Loneliness	a_3_	0.269	0.2960	0.041	0.189, 0.351	0.0005	R^2^ = 0.101 F(4,469) = 13.172 *p* = 0.0005
**Direct Effect *X* Frailty to Y Anxiety**	**c’**	**2.159**	**0.2068**	**0.439**	**1.2949, 3.024**	**0.0005**	**R^2^ = 0.228 F(4,469)=34.673 *p* = 0.0005**
M_1_ Care time to Y Anxiety	b_1_	0.676	0.1284	0.215	0.254, 1.099	0.0017	
M_2_ Financial difficulty to Y Anxiety	b_2_	2.673	0.1992	0.5767	1.539, 3.806	0.0005	
M_3_ Social loneliness to Y Anxiety	b_3_	2.756	0.2407	0.454	1.865, 3.647	0.0005	
Indirect Effects of X on Y							
**Total Indirect effects of X (a_1_ + b_1_ + a_2_ + b_2_ + a_3_ + b_3_)**		**1.743**		**0.289**	**1.226, 2.351**	**Sig**	
Care time (a_1_ + b_1_)		0.386		0.151	0.119, 0.716	Sig	
Financial difficulty (a_2_ + b_2_)		0.614		0.178	0.297, 0.986	Sig	
Social loneliness (a_3_ + b_3_)		0.744		0.177	0.429, 1.125	Sig	

**Table 4 ijerph-19-08636-t004:** Serial mediators in the relationship between caregiver frailty and anxiety.

Model Pathways		Coeff. β	Stan. Coeff	SE	95% CI	*p*	Model Summary
**Total effect of X on Y**	**c**	**3.903**		**0.439**	**3.040, 4.765**	**0.0005**	
***X* Frailty to Y Anxiety**	**c’**	**2.159**	**0.2068**	**0.439**	**1.295,3.024**	**0.0005**	
*X* Frailty to M1 Care work	a_1_	0.570	0.2879	0.091	0.393, 0.748	**0.0005**	R^2^ = 0.077 F(4,470) = 9.838 *p* = 0.0005
*X* Frailty to M_2_ Financial difficulty	a_2_	0.183	0.2143	0.034	0.117, 0.249	**0.0005**	R^2^ = 0.246 F(5,468) = 30.498 *p* = 0.0005
M_1_ Care time to M_2_ Financial difficulty	d_1,2_	0.110	0.2811	0.017	0.078, 0.143	**0.0005**	
X Frailty to M_3_ Social loneliness	a_3_	0.218	0.2389	0.044	0.132, 0.304	**0.0005**	**R^2^ = 0.123 F(6,467) = 10.874 *p* = 0.0005**
M1 Care time to M_3_ Social loneliness	d_1,3_	0.035	0.0768	0.022	−0.008, 078	0.1068	
M_2_ Financial difficulty to M_3_ Social Support	d_2,4_	0.139	0.1183	0.059	0.024, 0.254	**0.0182**	
**Direct Effect *X* Frailty to Y Anxiety**	**c’**	**2.159**	**0.2068**	**0.439**	**1.295, 3.024**	**0.0005**	R^2^ = 0.359 F(7,466) = 37.227 *p* = 0.0005
M_1_ Care time to Y Anxiety	b_1_	0.6764	0.1284	0.215	0.254, 1.099	**0.0017**	
M_2_ Financial difficulty to Y Anxiety	b_2_	2.673	0.1192	0.577	1.539, 3.806	**0.0005**	
M_3_ Social loneliness to Y Anxiety	b_3_	2.756	0.2407	0.454	1.865, 3.67	**0.0005**	
**Indirect effects of X on Y**							
**Total Indirect effects of X (a_1_ + b_1_ + a_2_ + b_2_ + a_3_ + b_3_)**		**1.743**	0.1670	**0.297**	**1.209, 2.369**	**Sig.**	
X Frailty to M1 Care time to Y Anxiety		0.386	0.0370	0.151	0.119, 0.707	**Sig.**	
X Frailty to M2 Financial Difficulty to Y Anxiety		0.446	0.0427	0.142	0.203, 0.759	**Sig.**	
X Frailty to M3 Social loneliness to Y Anxiety		0.600	0.0575	0.170	0.306, 0.976	**Sig.**	
X Frailty to M3 Social loneliness to Y Anxiety		0.168	0.0161	0.061	0.069, 0.307	**Sig.**	
X Frailty to M1 Care time to M3 Social loneliness to Y Anxiety		0.056	0.0053	0.036	−0.011, 0.131	Not (includes 0)	
X Frailty to M2 Financial Difficulty to M3 Social loneliness to Y Anxiety		0.064	0.0061	0.302	0.014, 0.129	**Sig.**	
X Frailty to M1 Care time to M2 Financial to M3 Social loneliness to Y Anxiety		0.024	0.0023	0.114	0.005, 049	**Sig.**	

## Data Availability

Data belong to the survey Impact of COVID-19 on Family Caregivers in Alberta and the data are owned by Jasneet Parmar the principal investigator. People interested in obtaining the data set can contact Sharon Anderson, research coordinator of the study. Email sdanders@ualberta.ca.

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
