# Peer review of "Family Caregiving during the COVID-19 Pandemic in Canada: A Mediation Analysis"

_ijerph, 2022, doi:10.3390/ijerph19148636_

Round 1

Reviewer 1 Report

Keywords: family caregiver 1; anxiety 2; frailty 3; COVID-10 4; mediation analysis 5; loneliness 6.

The keywords must be without numbers. Besides, not all of them should be in the paper's title.

1. Introduction

The introduction reviews 47 articles. The authors in the first five references refer to books, not journals, in which researchers investigated similar problems. Meanwhile, the authors present the list of references not according to the Journal's requirements. For example, they must download a few articles and prepare the list according to them. There are some examples of the latest published article:

1. Jang, Y.; Chiriboga, D.A.; Small, B.J. Perceived discrimination and psychological wellbeing: The mediating and moderating role of sense of control. Int. J. Aging Hum. Dev. 2008, 66, 213–227.

2. Cerci, P.A.; Dumludag, D. Life satisfaction and job satisfaction among university faculty: The impact of working conditions, academic performance, and relative income. Soc. Indic. Res. 2019, 144, 785–806.

3. McGuire, S.L.; Klein, D.A.; Chen, S. Ageism revisited: A study measuring ageism in East Tennessee, USA. J. Nurs. Health Sci. 2008, 10, 11–16.

4. Harris, K.; Krygsman, S.; Waschenko, J.; Laliberte Rudman, D.; Pruchno, R. Ageism and the older worker: A scoping review. Gerontologist 2018, 58, E1–E14.

5. Castaneda, M.; Isgro, K. Mothers in Academia; Columbia University Press: New York, NY, USA, 2013; pp. 1–188.

6. El-Alayli, A.; Hansen-Brown, A.; Ceynar, A. Dancing backwards in high heels: Female professors experience more work demands and special favor requests, particularly from academically entitled students. Sex Roles 2018, 79, 136–150.

9. Bureau of Labor Statistics. Employment-Population Ratio and Labor Force Participation Rate by Age. Available online: https://www.bls.gov/opub/ted/2017/employment-population-ratio-and-labor-force-participation-rate-by-age.htm (accessed on 28 April 2022).

10. Acosta, M.J.; Castillo-Sánchez, G.; Garcia-Zapirain, B.; De la Torre Diez, I.; Franco-Martín, M. Sentiment Analysis Techniques Applied to Raw-Text Data from a Csq-8 Questionnaire about Mindfulness in Times of COVID-19 to Improve Strategy Generation. Int. J. Environ. Res. Public Health 2021, 18, 6408.

11. Attiq, S.; Chau, K.Y.; Bashir, S.; Habib, M.D.; Azam, R.I.; Wong, W.K. Sustainability of household food waste reduction: A fresh insight on youth's emotional and cognitive behaviors. Int. J. Environ. Res. Public Health 2021, 18, 7013.

12. Attiq, S.; Chu, A.M.; Azam, R.I.; Wong, W.K.; Mumtaz, S. Antecedents of Consumer Food Waste Reduction Behavior: Psychological and Financial Concerns through the Lens of the Theory of Interpersonal Behavior. Int. J. Environ. Res. Public Health 2021, 18, 12457.

13. Cázares-Manríquez, M.A.; Camargo-Wilson, C.; Vardasca, R.; García-Alcaraz, J.L.; Olguín-Tiznado, J.E.; López-Barreras, J.A.; García-Rivera, B.R. Quantitative Models for Prediction of Cumulative Trauma Disorders Applied to the Maquiladora Industry. Int. J. Environ. Res. Public Health 2021, 18, 3830.

There are some examples with missing information about references in the presented paper:

12. Kumagai, N. "Distinct Impacts of High Intensity Caregiving on Caregivers' Mental Health and Continuation of Caregiving." Health Economics Review 7, no. 1 (2017): --> The authors must to add article number: Kumagai, N. "Distinct impacts of high intensity caregiving on caregivers' mental health and continuation of caregiving." Health Econ Rev 7, 15 (2017). https://doi.org/10.1186/s13561-017-0151-9

16. Barbosa, F., G. Voss, and A. Delerue Matos. "Health Impact of Providing Informal Care in Portugal." BMC Geriatrics 20, no. 1 (2020). --> Barbosa, F., G. Voss, and A. Delerue Matos. "Health impact of providing informal care in Portugal." BMC Geriatrics 20 (2020): 440. https://doi.org/10.1186/s12877-020-01841-z

18. Zhu, X. R., Z. R. Zhu, L. X. Wang, T. Zhao, and X. Han. "Prevalence and Risk Factors for Depression and Anxiety in Adult Patients with Epilepsy: Caregivers' Anxiety and Place of Residence Do Mater." Epilepsy and Behavior 129 (2022). --> Zhu, X. R., Z. R. Zhu, L. X. Wang, T. Zhao, and X. Han.. "Prevalence and risk factors for depression and anxiety in adult patients with epilepsy: Caregivers' anxiety and place of residence do mater." Epilepsy & Behavior 129 (2022): 108628.

31. Stewart, M., A. Barnfather, A. Neufeld, S. Warren, N. Letourneau, and L. L. Liu. "Accessible Support for Family Caregivers of Seniors with Chronic Conditions: From Isolation to Inclusion." Canadian Journal on Aging-Revue Canadienne Du Vieillissement 25, no. 2 (2006): 179-92. --> 179-192. https://doi.org/10.1353/cja.2006.0041

35. Alessi, J., G. B. de Oliveira, G. Feiden, B. D. Schaan, and G. H. Telo. "Caring for Caregivers: The Impact of the Covid-19 Pandemic on Those Responsible for Children and Adolescents with Type 1 Diabetes." Scientific Reports 11, no. 1 (2021). --> 11, no. 1 (2021): 6812. https://doi.org/10.1038/s41598-021-85874-3

36. Anderson, S., J. Parmar, B. Dobbs, and P. G. J. Tian. "A Tale of Two Solitudes: Loneliness and Anxiety of Family Caregivers Caring in Community Homes and Congregate Care." International Journal of Environmental Research and Public Health 18, no. 485 19 (2021). --> 18, no. 19 (2021): 10010. https://doi.org/10.3390/ijerph181910010

40. Budnick, A., C. Hering, S. Eggert, C. Teubner, R. Suhr, A. Kuhlmey, and P. Gellert. "Informal Caregivers During the Covid-19 Pandemic Perceive Additional Burden: Findings from an Ad-Hoc Survey in Germany." Bmc Health Services Research 21, no. 1 (2021). --> 21, no. 1 (2021): 353. https://doi.org/10.1186/s12913-021-06359-7

41. Dhiman, S., P. K. Sahu, W. R. Reed, G. S. Ganesh, R. K. Goyal, and S. Jain. "Impact of Covid-19 Outbreak on Mental Health and Perceived Strain among Caregivers Tending Children with Special Needs." Research in Developmental Disabilities 107 (2020). --> "Impact of COVID-19 outbreak on mental health and perceived strain among caregivers tending children with special needs." Research in Developmental Disabilities 107 (2020): 103790. https://doi.org/10.1016/j.ridd.2020.103790

42. Greenberg, N. E., A. Wallick, and L. M. Brown. "Impact of Covid-19 Pandemic Restrictions on Community-Dwelling Caregivers and Persons with Dementia." Psychological Trauma: Theory, Research, Practice, and Policy (2020). --> "Impact of COVID-19 pandemic restrictions on community-dwelling caregivers and persons with dementia." Psychological Trauma: Theory, Research, Practice, and Policy 12, no. S1 (2020): S220–S221. https://doi.org/10.1037/tra0000793.

45. Sinn, C. L. J., H. Sultan, L. A. Turcotte, C. McArthur, and J. P. Hirdes. "Patterns of Home Care Assessment and Service Provision before and During the Covid-19 Pandemic in Ontario, Canada." PLoS ONE 17, no. 3 March (2022). --> "Patterns of home care assessment and service provision before and during the COVID-19 pandemic in Ontario, Canada." PloS ONE 17, no. 3 (2022): e0266160. https://doi.org/10.1371/journal.pone.0266160

46. Anderson, S., J. Parmar, B. Dobbs, and P.G.J. Tian. "A Tale of Two Solitudes: Loneliness and Anxiety of Family Caregivers Caring in Community Homes and Congregate Care." Int. J. Environ. Res. Public Health 18 (2021). --> "A tale of two solitudes: Loneliness and anxiety of family caregivers caring in community homes and congregate care." Int. J. Environ. Res. Public Health 18, no. 19 (2021): 10010. https://doi.org/10.3390/ijerph181910010

55. Flaatten, H., B. Guidet, F. H. Andersen, A. Artigas, M. Cecconi, A. Boumendil, M. Elhadi, J. Fjølner, M. Joannidis, C. Jung, S. Leaver, B. Marsh, R. Moreno, S. Oeyen, Y. Nalapko, J. C. Schefold, W. Szczeklik, S. Walther, X. Watson, T. Zafeiridis, D. W. de Lange, and V. I. P. Study Group the. "Reliability of the Clinical Frailty Scale in Very Elderly Icu Patients: A Prospective 529 European Study." Annals of Intensive Care 11, no. 1 (2021). --> Flaatten, H., B. Guidet, F. H. Andersen, A. Artigas, M. Cecconi, A. Boumendil, M. Elhadi, J. Fjølner, M. Joannidis, C. Jung, S. Leaver, B. Marsh, R. Moreno, S. Oeyen, Y. Nalapko, J. C. Schefold, W. Szczeklik, S. Walther, X. Watson, T. Zafeiridis, D. W. de Lange. "Reliability of the Clinical Frailty Scale in very elderly ICU patients: a prospective European study." Annals of intensive care 11, no. 1 (2021): 22. https://doi.org/10.1186/s13613-021-00815-7

And others.

The authors must carefully check all the presented references and provide all information about them according to the Journal's requirements.

Discussion:

In this analysis, we were interested in the factors related to family caregiver anxiety and in hypothesizing how those effects occur. In regression analysis after controlling for gender and age, social loneliness (β= .245), frailty (β=.199), financial difficulty (β=.196), care time (β=.143), and navigation confidence (β=.131) were all significant.

-->The authors do not provide a strong background on why they selected this model to solve the problem. Besides, they do not show why this model to solve this problem is better than others. They do not show that researcher-developed many models to solve similar problems.

--> The conclusions must highlight the novelty of the research model and new findings. The authors do not provide conclusions.

Author Response

Thank you for your very through review. We really appreciate all the time that a review takes.  The changes are highlighted in red in the text. 

    1. The keywords must be without numbers. Besides, not all of them should be in the paper's title.

    Yes, We have removed the numbers and some of the redundancy.

    1. The authors must carefully check all the presented references and provide all information about them according to the Journal's requirements.

    Thank you. We really appreciate your very thorough guidance on the references.

    1. Discussion: In this analysis, we were interested in the factors related to family caregiver anxiety and in hypothesizing how those effects occur. In regression analysis after controlling for gender and age, social loneliness (β= .245), frailty (β=.199), financial difficulty (β=.196), care time (β=.143), and navigation confidence (β=.131) were all significant.

    The authors do not provide a strong background on why they selected this model to solve the problem. Besides, they do not show why this model to solve this problem is better than others. They do not show that researcher-developed many models to solve similar problems.

    Thank you We revised the introduction to make the theoretical background clearer. Our regression analysis showed the strongest link between frailty and anxiety which was not a surprise as caregivers’ health is often the strongest predictor of anxiety, distress, or burden. We were interested if mediation analysis might provide evidence similar to Smagula and colleagues’ [30] conclusions that care work might moderate the effect of caregivers’ frailty on anxiety.  As this is cross-sectional survey research, without experimental manipulation we are only theoretically testing our model. In the future, we are going to explore the link between caregiver health/ frailty and stress by examining some of the available longitudinal data. Often it is assumed that caregivers’ health declines from caring.  However, one review [1] and one meta-analysis [2] are questioning this.

    1. The conclusions must highlight the novelty of the research model and new findings. The authors do not provide conclusions.

    Thank you. Reviewer # 3 also suggested that we strengthen the conclusion by highlighting alignment with our hypothesis. We have added

    Our findings provide some insight into the relationship between caregivers’ health, measured as frailty, and their anxiety. Frailty levels were directly related to levels of caregivers’ anxiety and also indirectly moderated through weekly hours of care work, financial distress, and social loneliness.

    1. Roth, D.L.; Fredman, L.; Haley, W.E. Informal caregiving and its impact on health: A reappraisal from population-based studies. Gerontologist 2015, 55, 309-319, doi:10.1093/geront/gnu177.
    2. Mehri, N.; Kinney, J.; Brown, S.; Rajabi Rostami, M. Informal Caregiving and All-Cause Mortality: A Meta-Analysis of Longitudinal Population-Based Studies. Journal of Applied Gerontology 2021, 40, 781-791, doi:10.1177/0733464819893603.

Reviewer 2 Report

The study focuses on the little-covered problem "of the link between caregiver frailty, weekly care hours, perceptions of financial difficulty and social support and anxiety" (l. 93 - 94). It appropriately updates this topic in the COVID-19 pandemic context. Contributes to the knowledge of factors, "that moderate the connection between caregiver health and anxiety" (l. 91), but is "interested not only in the factors related to family caregiver anxiety, but also in hypothesizing how those effects occur" (l. 277).

The focus of this research is on "frailty" understood as "decline in functioning across multiple physiological symptoms" (l. 297). The authors formulate their hypotheses precisely and provide relevant, substantiated answers to research questions. They demonstrate research skills, and use several statistical methods, functions and calculations. In particular, the precise final analysis and discussion should be highlighted. They confronted their findings with many similar works to date and made relevant conclusions. They also achieved the desired overlap in practice, formulating proposals and recommendations for social practice in connection with their findings.

Small comment: The presented research was set in the Canadian environment and this should be emphasized in the introduction, where it is stated that "current health systems depend on family caregivers to provide 80-90% of the day-to-day assistance and care management required by care receivers living in the community ”(l. 37 - 38) etc. However, these are not globally valid data, so it would be appropriate to give them only as a (Canadian) example.

Author Response

relevant to the Canadian context.  We have added Canada to the title, Family Caregiving during the COVID-19 Pandemic in Canada: A Mediation Analysis  and in the introduction.  The changes are highlighted in red.

Reviewer 3 Report

Dear authors,

you have done extensive research, but with certain limitations. Still, I think your work is helpful.

It seems to me that the tone of presenting the results of your work is lost during its writing and is completely lost at the very end, ie in the conclusion.

Therefore, I advise you to first present in the "Conclusion" section the specific conclusions you have reached in relation to the hypotheses set out in your study. State them under ordinal numbers as well as hypotheses.

After that, make some suggestions so that the results of your research can help policymakers and healthcare providers.

Finally, since you point out the limitations of your study, in the "Conclusion", be sure to state its advantages, special features, and its contribution to the health system.

Author Response

Thank you for this lovely review and the excellent suggestions. We are very aware of the time that each review takes.  I have indented my reply in the text below and the changes are highlighted in red in the text. As requested by another reviewer, we re-wrote the introduction to make the theoretical background clearer.  

Dear Authors,

You have done extensive research, but with certain limitations. Still, I think your work is helpful.

It seems to me that the tone of presenting the results of your work is lost during its writing and is completely lost at the very end, ie in the conclusion.

Therefore, I advise you to first present in the "Conclusion" section the specific conclusions you have reached in relation to the hypotheses set out in your study. State them under ordinal numbers as well as hypotheses.

After that, make some suggestions so that the results of your research can help policymakers and healthcare providers.

Thank you very much for the compliment on the “extensive research” and your excellent suggestions. Reviewer 1 also suggested strengthening the conclusion.  We have added “Our findings provide some insight into the relationship between caregivers’ health, measured as frailty, and their anxiety. Frailty levels were directly related to levels of caregivers’ anxiety and also indirectly moderated through weekly hours of care work, financial distress, and social loneliness.”  

Finally, since you point out the limitations of your study, in the "Conclusion", be sure to state its advantages, special features, and its contribution to the health system.

Thank you. I  always find it difficult to comment on the study strengths. We added strengths after the limitations.,

 We used regression analysis and validated scales to examine the factors related to anxiety and then used mediation analysis to test hypotheses about the processes by which giving care can cause anxiety. Typically anxiety rises as care responsibilities, exhaustion, and worry increase [69], and in this study, moderation analysis confirmed that increased weekly care hours, financial distress, and social loneliness were significant. While anxiety is a known risk factor for poor health[69], mediation analysis demonstrated that frailty also increases anxiety directly and indirectly. This implies that healthcare providers should routinely assess family caregivers’ frailty and distress.       

Reviewer 4 Report

In this article, the authors included 474 caregivers after eighteen months of the COVID-19 pandemic to explore the link between caregiver frailty, weekly care hours, perceptions of financial difficulty and social support and anxiety. Besides, the authors pointed out the shortcomings of the existing study. Several comments are offered for authors’ consideration.

1.     The keywords may have a mistake: COVID-10.

2.     In section 2.1.1, the description of number of hours should also indicate “weekly”.

3.     There was a mistake about serial number of the manuscript.

4.     In practical research, the formulation of mediation variables needs theoretical basis or support, so the necessity of some variables should be described more.

5.     Why the authors chose Andrew Hayes PROCESS moderation analysis, because of its limitations, is it the most appropriate analysis? This section of the discussion deserves more explanation.

Author Response

Thank you for your excellent review. We know how long these reviews take and we really appreciate that reviewers take the time to make them better.  The changes are highlighted in red in the text. 

  1. The keywords may have a mistake: COVID-10.

Thank you My goodness we should have noticed this! We also removed the numbers and added relevant keywords.

  1. In section 2.1.1, the description of number of hours should also indicate “weekly”.

We appreciate this.  We have added weekly in several places

  • Weekly Care work.

Family caregivers, who were caring before COVID-19 were asked whether care time increased, remained stable, or decreased. We assessed the number of hours devoted to care time weekly

  1. There was a mistake about serial number of the manuscript.

I think the serial number of the manuscript is given if and when the article is accepted. 

  1. In practical research, the formulation of mediation variables needs theoretical basis or support, so the necessity of some variables should be described more.

We have rewritten the introduction to make the theoretical basis for choosing these variables clearer. Our regression analysis showed the strongest link between frailty and anxiety which was not a surprise as caregivers’ health is often the strongest predictor of anxiety, distress, or burden. We were interested if mediation analysis might provide evidence similar to Smagula and colleagues’ [30] conclusions that care work might moderate the effect of caregivers’ frailty on anxiety.  As this is cross-sectional survey research, without experimental manipulation we are only theoretically testing our model. In the future, we are going to explore the link between caregiver health/ frailty and stress by examining some of the available longitudinal data. Often it is assumed that caregivers’ health declines from caring.  However, one review [1] and one meta-analysis [2] are questioning this.

  1. Why the authors chose Andrew Hayes PROCESS moderation analysis, because of its limitations, is it the most appropriate analysis? This section of the discussion deserves more explanation.

We could have used Process medication analysis or Structural equation modeling. They produce similar results [3].  We were using SPSS and Process is available for SPSS.  As above, this is really an exploratory analysis.

  1. Roth, D.L.; Fredman, L.; Haley, W.E. Informal caregiving and its impact on health: A reappraisal from population-based studies. Gerontologist 2015, 55, 309-319, doi:10.1093/geront/gnu177.
  2. Mehri, N.; Kinney, J.; Brown, S.; Rajabi Rostami, M. Informal Caregiving and All-Cause Mortality: A Meta-Analysis of Longitudinal Population-Based Studies. Journal of Applied Gerontology 2021, 40, 781-791, doi:10.1177/0733464819893603.
  3. Hayes, A.F.; Montoya, A.K.; Rockwood, N.J. The analysis of mechanisms and their contingencies: PROCESS versus structural equation modeling. Australasian Marketing Journal 2017, 25, 76-81, doi:10.1016/j.ausmj.2017.02.001.

Reviewer 5 Report

This appears to be a competently carried out study, both conceptually and empirically. My only comments is the the participants' characteristics that are presented as results should appear in the methods section.

Author Response

Thank you so much for your complementary review. We really appreciated the time that each review takes.  We have made changes in red in the text. 

This appears to be a competently carried out study, both conceptually and empirically. My only comments is the participants' characteristics that are presented as results should appear in the methods section.

Thank you for the compliment about conceptual and empirical competence. We have been researching and working with family caregivers since 2014 and want to start moving beyond associations. I have been working with PROCESS and structural equation modeling and recently took a mediation and moderation analysis course from Andrew Hayes.  This study is exploratory with cross-sectional data. There is lots more research to do.

There are times when participants’ characteristics are in methods, however, yours is the 5th review on this paper and moving them to methods was not requested by other reviewers. My PhD supervisor also always requested them at the beginning of the results section on the basis that “You have not yet recruited your participants. That is a result.”

Round 2

Reviewer 1 Report

... Frailty is a ... 317 line

public health measure that indicates increased risk of poor health[57-59] [57-59]. Rock- ... 318 line ...

There is no need to cite twice the same references in one sentence.

Author Response

... Frailty is a ... 317 line

public health measure that indicates increased risk of poor health[57-59] [57-59]. Rock- ... 318 line ...

There is no need to cite twice the same references in one sentence.

Thank you for your careful reviews. We reiterated what we said after the first review, we really appreciate the time that it takes to do a through review.

We have corrected this.

In this study we used a self-report version of the Clinical Frailty Scale [CFS] [50,51] initially used in an assessment study of the impacts of frailty assessment and social prescribing [52]. The CFS was originally validated for face-to-face screening by a trained healthy provider, but adaptations are permitted. We used Rasiah’s [52] nine questions to assess caregiver’s frailty (See CFS Scale in Appendix B). 

Reviewer 4 Report

I think this manuscript could be accepted in present form.

Author Response

I think this manuscript could be accepted in present form.

Thank you. For the two reviews! As I said previously, we really appreciate the time that reviewing takes.